# Recyclable soft photonic crystal film with overall improved circularly polarized luminescence

Yonghong Shi[1,2], Jianlei Han[1], Chengxi Li[1,2], Tonghan Zhao [1], Xue Jin[1] & Pengfei Duan [1,2] ✉

Existing circularly polarized luminescence materials can hardly satisfy the requirements of both large luminescence dissymmetry factor and high luminescent quantum yield, which hinders their practical applications. Here, we present a soft photonic crystal film embedded with chiral nanopores that possesses excellent circularly polarized luminescence performance with a high luminescence dissymmetry factor as well as a large luminescent quantum yield when loaded with various luminescent dyes. Benefitting from the retention of chiral nanopores imprinted from a chiral liquid crystal arrangement, the chiral soft photonic crystal film can not only endow dyes with chiral properties, but also effectively avoid severe aggregation of guest dye molecules. More importantly, the soft photonic crystal film can be recycled many times by loading and eluting guest dye molecules while retaining good stability as well as circularly polarized luminescence performance, enabling various applications, including smart windows, multi-color circularly polarized luminescence and anticounterfeiting.

Chiroptical materials with circularly polarized luminescence (CPL) hold great promise for a diverse range of applications across multiple fields, including 3D optical displays, optical data storage, photoelectric devices[1–8] and information encryption[9–11]. With regard to real applications of CPL materials, pursuing a large luminescence dissymmetry factor ($g_{lum}$) is of vital importance[12,13]. Over the years, photochemists have developed a variety of strategies for amplifying $g_{lum}$, which include but are not limited to molecular self-assembly[14–18], aggregation-induced emission (AIE)[19,20], energy transfer involving donor-acceptor systems[21,22], or localized surface plasmon resonance[23,24]. However, in most of the strategies, the $g_{lum}$ values are still in the range of $10^{-3}$–$10^{-1}$[25], which is far from practical applications. Therefore, there is still an intense demand for further amplifying the $g_{lum}$ value.

Chiral nematic liquid crystal (N*LC) have been studied as excellent host materials for realizing CPL-active materials with large $g_{lum}$ values[26–28]. Generally, the preferred approach for achieving an N*LC is to dope the chiral compounds into the an achiral nematic liquid crystal. The photonic bandgap (PBG) of an N*LC could be flexibly tuned by changing the proportion of chiral dopants. Achiral emitters could be endowed with CPL performance after coassembly with the N*LC. Additionally, the doped emitters can a show very large $g_{lum}$ in the N*LC system when the emission peaks are located within the PBG[29–34]. Although a large $g_{lum}$ can be obtained in the N*LC system, many problems are still encountered in the practical application of CPL materials, including the following: (1) liquid crystals have liquidity, particularly room-temperature liquid crystals, and need to be sealed in liquid crystal cells; (2) under stimulation by heat or external mechanical forces, the regular arrangement of an N*LC is easily disordered, which quenches the CPL emission; and (3) many organic luminescent dyes have low emission intensity in liquid crystal systems due to aggregation caused quenching (ACQ) effects; worse still, liquid crystal systems can quench most of the emission due to the strongly polar environment. Thus, to overcome the shortcomings and difficulties

[1]CAS Key Laboratory of Nanosystem and Hierarchical Fabrication, National Center for Nanoscience and Technology (NCNST), No. 11 ZhongGuanCun BeiYiTiao, 100190 Beijing, PR China. [2]University of Chinese Academy of Sciences, 100049 Beijing, PR China. ✉e-mail: duanpf@nanoctr.cn

listed above, reverse chiral soft photonic crystal (SPC) films provide an excellent solution. SPC films could be fabricated by using a chiral liquid crystal as a template, replicating its three-dimensional structure within a polymer, and then removing the chiral liquid crystal. Coles et al. achieved[35], for the first time, a porous cast film with chiral three-dimensional structure of blue phase by removing the remaining liquid crystal from the photopolymerized blue phase. The cast film can serve as a hard template for the fabrication of new materials.

Inspired by polymer templating of the bule phase[36,37], we speculate that the porous structure of an SPC films can effectively avoid severe aggregation of loaded dye molecules while retaining the three-dimensional texture of the chiral liquid crystal, enabling excellent chiroptical properties. Moreover, the excellent thermal stability of the SPC films can resist temperature disturbances. Overall, the SPC films can be used as a template to realize CPL materials with large $g_{lum}$ and high $\Phi$, which would serve as an ideal platform for practical applications. However, to date, there is no report about chiral SPC films in the research field of CPL. Therefore, a thorough investigation and understanding of the CPL properties of chiral SPC films is needed.

Here, we apply a chiral SPC films embedded with chiral nanopores as a recyclable and versatile chiral host matrix to produce a series of achiral guest dyes with a CPL property. The chiral SPC (SPC$^S$ or SPC$^R$) film can be fabricated by replicating the three-dimensional chiral structure of an N*LC through photopolymerization, followed by subsequent washing out of the N*LC component (Fig. 1). The obtained chiral SPC film is used as a versatile platform for loading various fluorescent dyes, endowing these dyes with good CPL activity. Three representative luminescent dyes, including ones with AIE (tetraphenylethylene, TPE) and ACQ (spiropyran, SP; perylene; coumarin 6, C6) behavior and one kind of dye that always exhibits highly efficient emission in any state or condition (9,10-diphenylanthracene, DPA), are thoroughly investigated after being loaded into chiral SPC films. AIE and ACQ dyes cannot simultaneously possess high $\Phi$ in the same medium. However, after being embedded into chiral SPC films, both large $g_{lum}$ and high $\Phi$ can be concurrently realized. It has been verified

that guest dyes can be orderly adsorbed in the nanopores of chiral SPC films, which can avoid severe aggregation, enabling better CPL performance with large $g_{lum}$ and high $\Phi$. Therefore, the dye-loaded chiral SPC film could be used for information encryption based on the better CPL performance in that the reflection and emission can be easily distinguished with the naked eye assisted by left- and right-handed circular polarization filters (L/R-CPFs). Additionally, the chiral SPC film can be recycled many times through immersion and removal of guest dye molecules while retaining good stability and CPL performance (Fig. 1). Based on its many advantages, we implement functional applications of the film in areas including smart windows, multi-color emission CPL, information encryption and anticounterfeiting. Therefore, as a general way to construct CPL-active materials with large $g_{lum}$ and $\Phi$, the approach presented here will provide more reliable alternative materials for the practical application and development of CPL-active materials.

## Results

### Chiral SPC film fabrication and characterization

Chiral liquid crystal imprinted nanopore SPC films were fabricated with the following chemical compositions: HTG135200/C6M/S5011 (or R5011)/trimethylolpropane triacrylate (TMPTA)/Irgacure 651 (I-651) = 100/50/0.9 ~ 2.4/7.8/2.5 (wt%) (the abbreviations and chemical structures of the chemicals are shown in Supplementary Table 1). Here, the chiral SPC film retains the nanopore imprints of the N*LC arrangement, where the direction of the chiral signal is determined by the chiral dopant S5011 (SPC$^S$) or R5011 (SPC$^R$). To fabricate chiral SPC films, the mixtures were injected into an LC cell, followed by exposure to UV-365 nm light (6W) for 30 min. After removal of the liquid crystal cell, the unpolymerized materials, including N*LC, polymeric monomer and initiator, were subsequently washed out by hexane, obtaining the chiral SPC film with an obvious reflection band (Supplementary Figs. 1 and 2a). Compared with the film before washing, the PBG of the SPC film exhibited a blueshift due to the pitch change caused by the absence of liquid crystal molecules[38]. The texture of the SPC$^S$ film

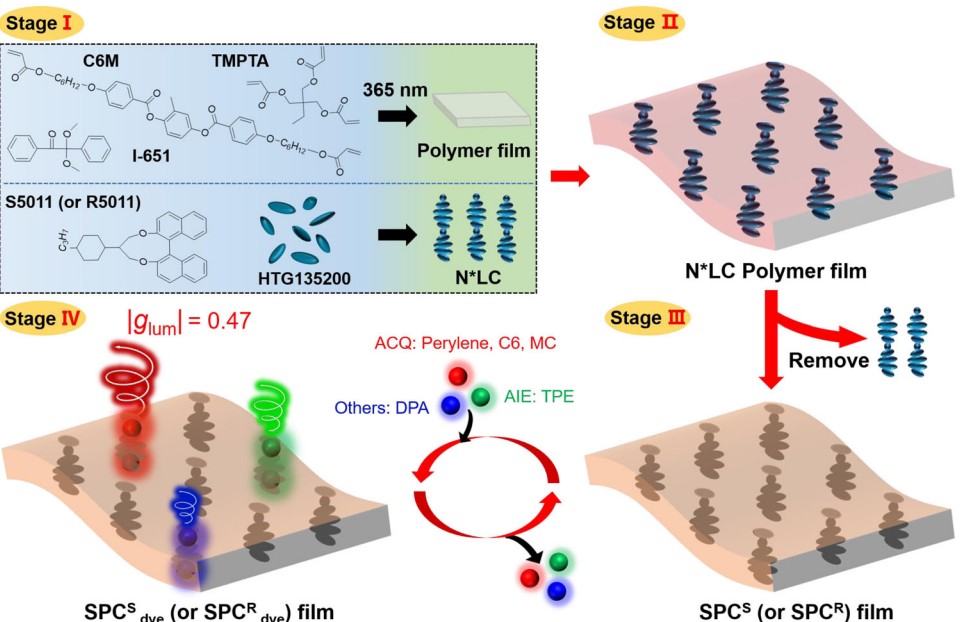

**Fig. 1 | Preparation of a chiral SPC (SPC$^S$ or SPC$^R$) film and general strategy for obtaining CPL materials.** Stage I: components of an N*LC polymer film. Stage II: mixture components were exposed to 365 nm for photopolymerization to form N*LC polymer film. Stage III: the N*LC polymer film was placed in hexane to wash out the liquid crystal, chiral dopant and residual monomer. SPC films embedded with chiral nanopores (SPC$^S$ or SPC$^R$) were obtained. Stage IV: three representative luminescent dyes, including those with AIE (TPE) and ACQ (SP, perylene, C6) behavior and one kind of dye that always exhibits highly efficient emission in any state or condition (DPA), showed large $g_{lum}$ and high $\Phi$ after loading into the SPC$^S$ or SPC$^R$ films (SPC$^S_{dye}$ or SPC$^R_{dye}$). Moreover, the chiral SPC films can be recycled many times through immersion and removal of guest dye molecules.

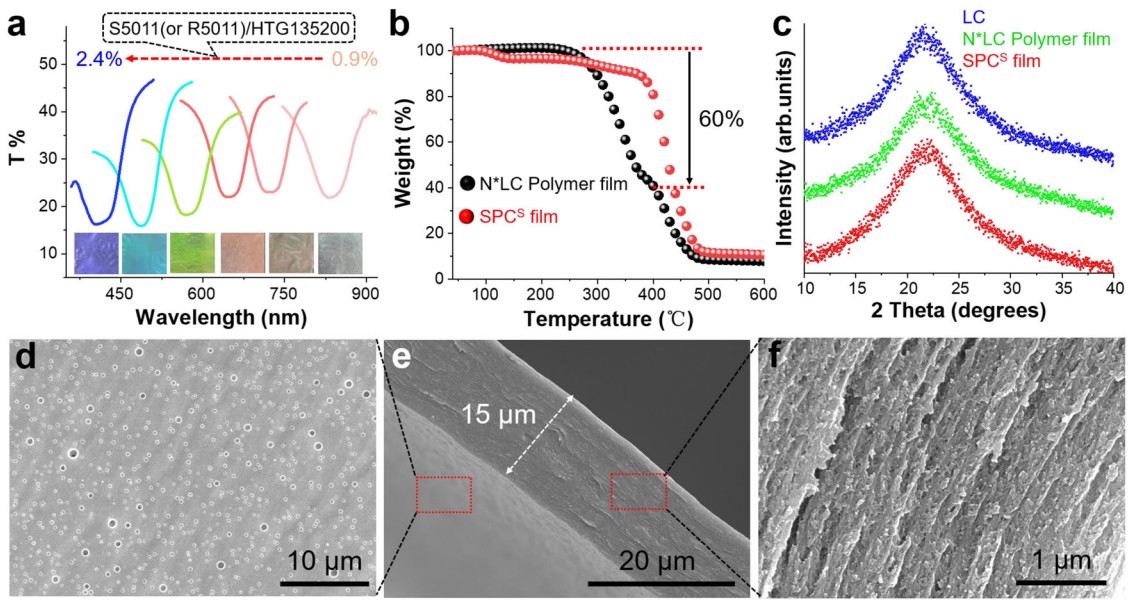

**Fig. 2 | Characteristics of the SPC$^S$ film. a** Transmittance spectra of SPC$^S$ films with different ratios of S5011/HTG135200 (wt%). The inset shows related images in natural light. **b** TGA curves indicating that ~60 wt% HTG135200 was completely removed from the N*LC polymer film. **c** XRD spectra of the N*LC (S5011, HTG135200), N*LC polymer and SPC$^S$ film. SEM image of an SPC$^S$ film (S5011/HTG135200 = 1.35 wt%), showing the structural features of exterior surfaces (**d**) and the cross section (**e, f**).

remained the same after removing the unpolymerized material (Supplementary Fig. 2b–d). Here, an SPC$^S$ film with a PBG range from the UV to near-infrared (NIR) region can be easily obtained by tuning the ratio of S5011 (or R5011)/HTG135200 (0.9 ~ 2.4 wt%). The corresponding SPC$^S$ film photographs are shown in the inset (Fig. 2a). Thermogravimetric analysis (TGA) further verified that the unpolymerized liquid crystal molecules and chiral molecules were completely removed after washing. TGA curves showed ~60 wt% weight loss in the original film, indicating that ~60 wt% nonreactive liquid crystal molecules were removed (Fig. 2b). Nearly the same weight loss (64.4%) was found after washing with hexane (Supplementary Fig. 3), which is essentially similar to the theoretical value (63.4% nonreactive materials). These data strongly confirm the complete removal of liquid crystal molecules. Meanwhile, removal of the nonreactive liquid crystal molecules led to shrinkage of the polymer network, shortening the area relative to the substrate surface by 34% (Supplementary Fig. 3). X-ray diffraction (XRD) results indicated that the SPC$^S$ film showed a diffraction pattern similar to those of the original nematic liquid crystal HTG135200 and liquid crystal molecule-loaded polymer film (Fig. 2c). This can be attributed to the chiral imprint structure imparted by the N*LC in the SPC$^S$ film.

The nanostructure of the SPC$^S$ film was studied by using scanning electron microscopy (SEM). Chiral nanopores with a size of less than 300 nm were observed on the exterior surface of the SPC$^S$ film (Fig. 2d). Meanwhile, we studied the internal structure of the SPC$^S$ film from the cross section obtained by SEM. From Fig. 2e, the thickness of the SPC$^S$ film was clearly 15 μm. Apparently, a large number of nanopores were found within the cross section of the SPC$^S$ film (Fig. 2f). To further determine the pore size, the chiral SPC film was analyzed by N$_2$ adsorption-desorption isotherms, and its nanopores were mainly distributed in the size range of 4–7 nm (Supplementary Fig. 4).

### Photophysical properties of the dye-loaded chiral SPC film
Considering that functional materials based on molecular isomer switches can achieve light-driven reversible changes in external properties such as color[39,40], shape[41] and fluorescence[42,43], which show great potential in the fields of sensors[44–46], actuators[47,48], soft systems[49] and anticounterfeiting systems[50,51], herein, an ACQ effect molecule,

photofluorochromic SP[52–54], was used as a representative material for constructing a switchable CPL-active SPC film. SP can convert to the colored and fluorescent merocyanine (MC) isomer upon irradiation with UV light and revert to the colorless SP isomer upon heating or visible light irradiation (Fig. 3). For the fabrication of the SP-loaded chiral SPC film (SPC$^S$ $_{SP}$), a chiral SPC film with a PBG at 670 nm was selected for soaking in SP solution (methanol, [SP] = 5 × 10$^{-4}$ mol l$^{-1}$) for 60 min, then washed and dried. It should be emphasized that various fluorescence dye has different solubility in methanol solution, and in order to quantify all dye, a chiral film soaked in dye solution (methanol, [dye] = 5 × 10$^{-4}$ mol l$^{-1}$) for 60 min (Supplementary Fig. 5). Upon exposure to UV-365 nm light, the SPC$^S$ $_{SP}$ film changed from a dark-yellow to a purple-red color (SPC$^S$ $_{MC}$) with red emission. Subsequently, the SPC$^S$ $_{MC}$ film returned to the SPC$^S$ $_{SP}$ state via heating or irradiation with visible light (465 nm LED light) (Fig. 3a). Photographs of each stage are shown in Fig. 3b under natural light (NL) and UV-365 nm light (UV), where the reflection and fluorescence intensity showed significant differences under the L-CPF and R-CPF (Fig. 3c). We also analyzed the effect of the SP molecules on the SPC$^S$ film by obtaining polarized optical microscopy (POM) images and transmittance spectra (Supplementary Figs. 6 and 7). The results indicated that the SPC$^S$ $_{SP}$ film showed similar planar texture and PBG position to the SPC$^S$ film. This can be explained by the loaded SP molecules not interfering with the structure of the SPC$^S$ film. In addition, from the SEM images of the external and internal nanostructures of the SPC$^S$ and SPC$^S$ $_{SP}$ films (Supplementary Fig. 8), we can see that the films exhibited similar nanostructures before and after loading the guest dye molecules.

The UV–vis absorption and fluorescence spectra showed that the SPC$^S$ $_{MC}$ film has an absorption peak at 570 nm with a dark-red emission located at 670 nm (Fig. 3d, $\lambda_{ex}$ = 365 nm, $\Phi$ = 22.3%, Supplementary Table 2). Notably, mirror-imaged CPL signals of SPC$^S$ $_{MC}$ and SPC$^R$ $_{MC}$ could be obtained (Fig. 3e), where the positive and negative signals of nanopores were determined by S5011 and R5011, respectively. The corresponding |$g_{lum}$| was 0.47 at 670 nm. Additionally, photoswitching reactions were repeatedly conducted by alternating irradiation of UV-365 nm and 465 nm LED light (Supplementary Fig. 9). There was no appreciable alteration of the |$g_{lum}$| value after ten cycles, indicating excellent reversibility and stability. Significantly, stable $g_{lum}$ value was

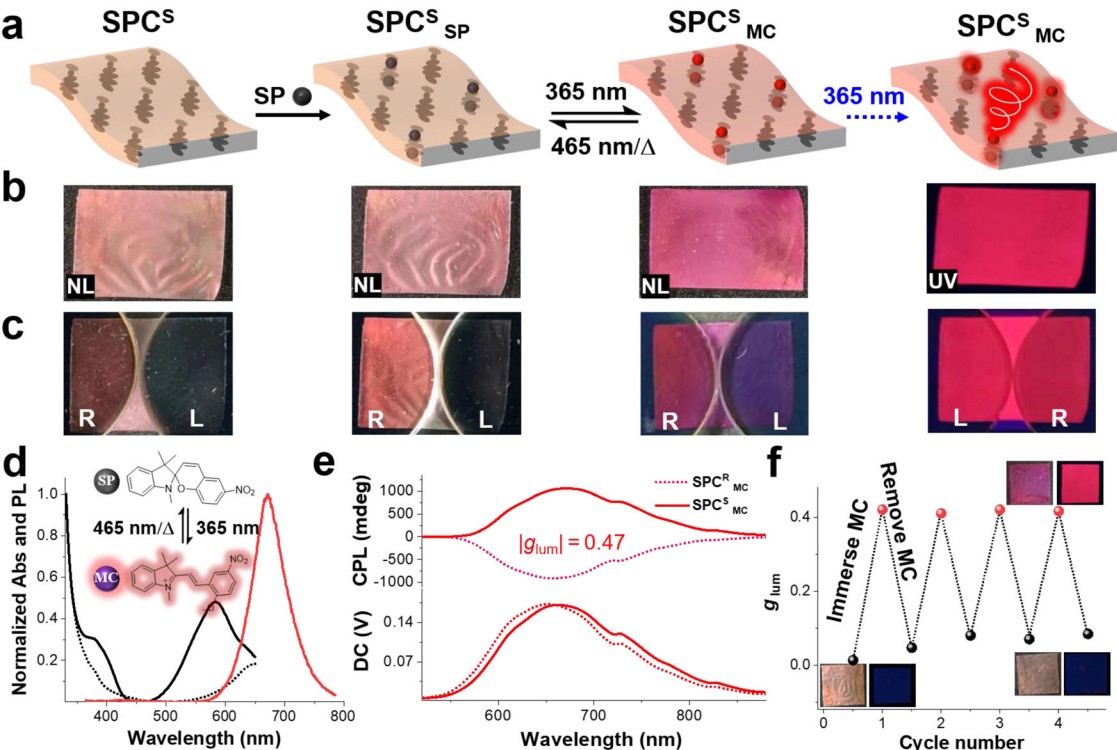

**Fig. 3 | Photophysical properties of the SPC$^S$ $_{MC}$ film. a** Schematic illustration of the fabrication processes of SPC$^S$ $_{MC}$. **b** Photographs of different stages under natural light (NL) and 365 nm (UV) light. **c** Corresponding images of (**b**) when the L-CPF and R-CPF were placed on top of the sample. **d** Normalized absorption and fluorescence of SPC$^S$ $_{SP}$ and SPC$^S$ $_{MC}$ films; the inset shows switching between the SP and MC isomers. **e** CPL spectra of SPC$^S$ $_{MC}$ and SPC$^R$ $_{MC}$ films excited at 360 nm. **f** The $g_{lum}$ value in the SPC$^S$ film sample was monitored at 670 nm during cyclic immersion into and removal of MC molecules, and the inset shows related images in natural light (left) and UV light (right). Schematic representation of experimental setup for fluorescence and CPL spectra is shown in Supplementary Fig. 12.

obtained after multiple soaking/removal cycles of the SP molecules, confirming that the SPC$^S$ film could be reused without damage (Fig. 3f). To verify the importance of the chiral nanopores in the SPC film, we have tested the permeation effect of dye molecules in normal polymer film without chiral nanopores. As shown in Supplementary Fig. 10, it is clearly observable that the amount of SP molecules entering the chiral nanopores of SPC$^S$ film is much more than the penetration of SP molecules in the normal polymer network. To clarify the ground state chirality of the chiral SPC film and MC isomer, the center position of the PBG of the chiral SPC film was tuned to 750 nm (eliminating the interference of the PBG). The SPC$^S$ and SPC$^R$ exhibit distinct mirror-image CD signals, which can be attributed to the chiral imprinted structures imparted by the chiral liquid crystal (Supplementary Fig. 11a). Notably, a new mirror signal centered at 570 nm (beyond PBG = 750 nm) was observed in the CD spectra of both SPC$^S$ $_{MC}$ and SPC$^R$ $_{MC}$ samples, strongly suggesting the chiral transfer from chirally imprinted structures to achiral MC dye (Supplementary Fig. 11b). To ensure that the observed signals are not influenced by linear dichroism or scattering, we conducted a rotation angle test on the samples. The $g_{CD}$ value remained unchanged, indicating that the influence of line deviation can be disregarded (Supplementary Fig. 11c, d).

## Generality of the strategy

Generally, the fluorescence properties of dyes with AIE and ACQ properties in the same medium are contradictory[55–58]. Interestingly, a chiral SPC film can be used as a versatile host platform for loading various fluorescent dyes, endowing guest dyes with large $g_{lum}$ and high $\Phi$. We selected some representative dyes with different properties, such as TPE (AIE effect), C6 and perylene (ACQ effect) and DPA (a typical dye that always shows highly efficient fluorescence in any condition and state). First, SPC$^S$ and SPC$^R$ films with suitable PBGs were selected for soaking in DPA and C6 solutions (methanol,

[DPA] = [C6] = 5 × 10$^{-4}$ mol l$^{-1}$) for 60 min and then dried to obtain SPC$^S$ $_{DPA}$, SPC$^R$ $_{DPA}$ SPC$^S$ $_{C6}$ and SPC$^R$ $_{C6}$ films (Fig. 4a and Supplementary Fig. 13). As expected, the obtained SPC$^S$ $_{DPA}$ ($\Phi$ = 99.4%) and SPC$^S$ $_{C6}$ ($\Phi$ = 75.2%) films both showed considerable fluorescence (Supplementary Fig. 14 and Table 2). Meanwhile, the SPC$^S$ $_{PDA}$ and SPC$^R$ $_{PDA}$ films showed mirror-image CPL signals with corresponding |$g_{lum}$| values of 0.43 ($\lambda_{em}$ = 430 nm). Similarly, the |$g_{lum}$| values of the SPC$^S$ $_{C6}$ and SPC$^R$ $_{C6}$ films reached 0.44 at 530 nm (Fig. 4b). It is reasonable that the chiral signal of the SPC film was determined by the chirality of the imprints of the N*LC nanopores. The reflection and fluorescence intensities of the SPC$^S$ $_{DPA}$ and SPC$^S$ $_{C6}$ films showed obvious distinctions under the R-CPF and L-CPF (Fig. 4c). The same planar texture of the chiral SPC film was observed by POM before and after loading the dye molecules (Supplementary Fig. 15). Likewise, loading the AIE molecule TPE and the ACQ molecule perylene into an SPC$^S$ film with a suitable PBG can produce SPC$^S$ $_{TPE}$ and SPC$^S$ $_{Perylene}$ films (methanol, [TPE] = [perylene] = 5 × 10$^{-4}$ mol l$^{-1}$). Interestingly, high fluorescence efficiency of SPC$^S$ $_{TPE}$ ($\Phi$ = 32.9%) and SPC$^S$ $_{Perylene}$ ($\Phi$ = 84.0%, Supplementary Fig. 14 and Table 2), as well as large |$g_{lum}$| values of 0.42 ($\lambda_{em}$ = 415 nm) and 0.45 ($\lambda_{em}$ = 456 nm, Supplementary Fig. 16), could be achieved. Thus, regardless of the loaded dyes, the obtained chiral SPC films all exhibited good CPL performance, indicating that the chiral nanopore imprint left by the N*LC can be preserved. To the best of our knowledge, CPL materials possessing large $g_{lum}$, high $\Phi$ and recyclability are very rare.

## Mechanism of chiral transfer and high fluorescence quantum yield

To get more insights on the chiral transfer from the chiral SPC film to achiral dye molecules, we selected C6-loaded chiral SPC$^S$ films as representative example for elucidation. When the C6 molecule was embedded into the SPC$^S$ films, the SPC$^S$ $_{MC}$ film exhibits a new CD

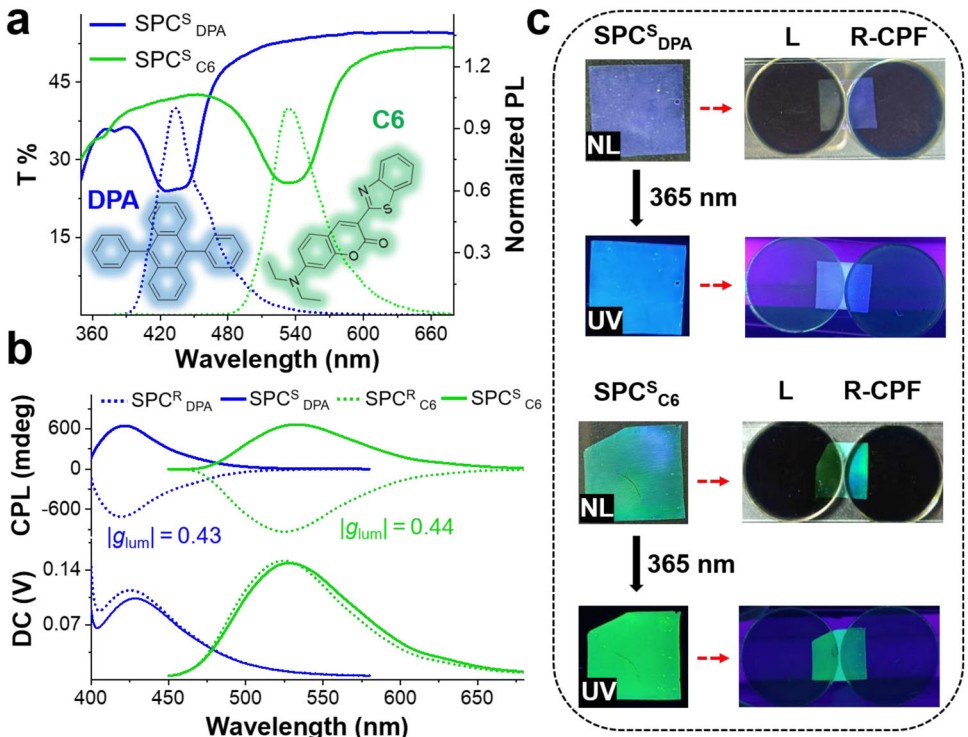

**Fig. 4 | CPL properties of the SPC$^S$ $_{DPA}$/SPC$^R$ $_{DPA}$ and SPC$^S$ $_{C6}$/SPC$^R$ $_{C6}$ films.**
**a** Transmittance and normalized emission spectra of SPC$^S$ $_{DPA}$ and SPC$^S$ $_{C6}$ films, $\lambda_{ex}$ = 360 nm; inset images show the related molecular structure. **b** CPL spectra of SPC$^S$ $_{DPA}$/SPC$^R$ $_{DPA}$ ($\lambda_{em}$ = 430 nm, $|g_{lum}|$ = 0.43) and SPC$^S$ $_{C6}$/SPC$^R$ $_{C6}$ ($\lambda_{em}$ = 530 nm, $|g_{lum}|$ = 0.44) films at an excitation wavelength of 360 nm. **c** Images of SPC$^S$ $_{DPA}$ and SPC$^S$ $_{C6}$ films under natural light (NL) and UV-365 nm light and with the L-CPF and R-CPF in natural light and UV light. SPC$^S$ $_{DPA}$: S5011/HTG135200 = 2.3 wt%, SPC$^S$ $_{C6}$: S5011/HTG135200 = 1.9 wt%.

signal in the 400–500 nm (Supplementary Fig. 17). Furthermore, we conducted investigations on the photophysical properties of SPC$^S$ films by immersing them in a dilute solution of C6 molecules at various concentrations (ranging from $10^{-6}$ to $5 \times 10^{-5}$ mol l$^{-1}$) for 10 min. As the concentration of C6 solution increased, both the absorption and emission of C6 molecules within the SPC$^S$ film also increased. This result suggests a strong correlation between the quantity of embedded dye molecules in the SPC$^S$ film and the concentration of the solution (Supplementary Fig. 18a, b). Interestingly, the position of the absorption spectra of SPC$^S$ $_{C6}$ did not exhibit a significant shift with an increasing amount of embedded C6 molecules. However, we observed a clear augmentation in the circular dichroism (CD) values of the SPC$^S$ $_{C6}$ film as the concentration of the C6 molecules enhanced (Supplementary Fig. 18c, d). Based on the experimental results mentioned above, we concluded that the SPC$^S$ $_{C6}$ film at low concentration displayed low CD values, possibly due to the limited adsorption of C6 molecules on the chiral nanopores imprint. This limited adsorption resulted in the absence of an orderly arrangement (Supplementary Fig. 18e). Conversely, as the concentration increased, the C6 molecules were more orderly adsorbed onto the chiral nanopores imprint of the SPC$^S$ film, leading to a stronger CD signal. Thus, the chirality of the guest dyes is closely associated with the loading amount, indicating that the induced chirality should be attributed to the supramolecular chirality of the dyes, rather than the selective reflection of the chiral nematic liquid crystal (N*LC). Analysis of the absorption and CD spectra of SPC$^S$ $_{C6}$ in dilute solutions (ranging from $10^{-6}$ to $5 \times 10^{-5}$ mol l$^{-1}$, Supplementary Fig. 18) enables us to infer that the embedded dye molecules are adsorbed in an ordered manner within the chiral nanopores imprint. This ordered adsorption of dye molecules on the chiral nanopores imprint limits their internal rotations, thereby leading to a higher fluorescence quantum yield, particularly in dye molecules with the AIE effect in the SPC$^R$ or SPC$^S$ films.

To further investigate the fluorescence quantum yield of SPC$^R$ or SPC$^S$ films following immersion in a highly concentrated dye solution, we chose C6 molecules with the ACQ effect for the experiments. The CD signal of the SPC$^S$ $_{C6}$ film significantly increased with longer immersion time in the supersaturated solutions. When the immersion time reached ~14 h, the dye molecules in the SPC$^S$ film reached saturation (Supplementary Fig. 19a, b). Notably, throughout the entire immersion process, the absorption and emission spectra of SPC$^S$ $_{C6}$ did not show any noticeable shifts (Supplementary Fig. 19c, d). We hypothesize that when the chiral nanopores imprint was completely filled with dye molecules, the solvent's pull prevented significant aggregation among the dye molecules, thus keeping them in solution (Supplementary Fig. 20a, b).

To further validate our hypothesis, we examined the dispersion state of the guest dyes by testing the UV–vis and emission spectra. The absorption spectra of the SPC$^S$ $_{C6}$ film in supersaturated solutions for 24 h displayed a slight red shift compared to the C6 solution state, indicating minimal aggregation behavior in the film (Supplementary Fig. 20c). Corresponding XRD results also confirmed the absence of severe aggregates of C6 molecules in the SPC$^S$ film (Supplementary Fig. 20d). Similar results were observed for the absorption spectra of perylene molecules in the SPC$^S$ film (Supplementary Fig. 20e). Therefore, it is evident that dye molecules with both ACQ and AIE effects exhibit high fluorescence quantum yields in the chiral SPC film. This can be attributed to the orderly adsorption of dye molecules within the nanopores, which restricts intramolecular rotation. Additionally, the influence of the solvent within the imprinted nanopores prevents significant aggregation among the dye molecules.

**Applications of the chiral SPC film**
Inspired by the excellent CPL performance of the chiral SPC film, we fabricated a white-light emissive chiral SPC film. First, the soaking solution was made of three different dyes (DPA/C6/MC, optimized

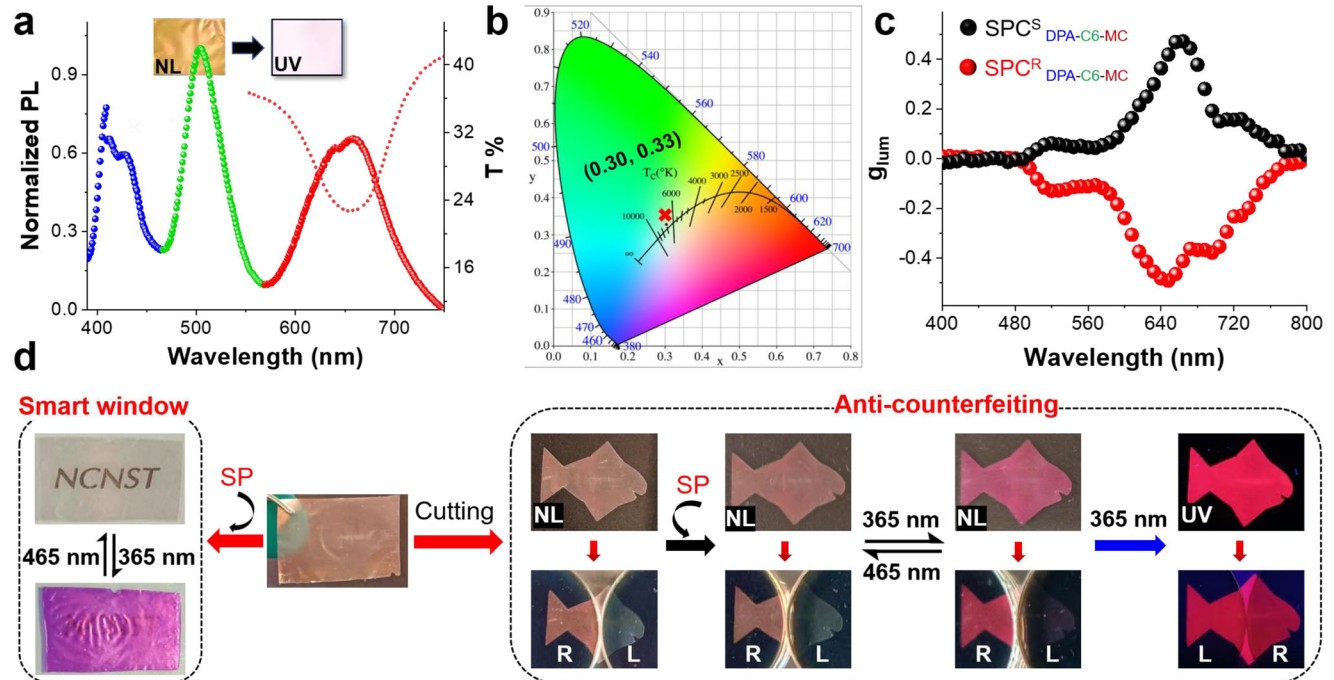

**Fig. 5 | Chiral SPC film applications. a** Fluorescence and transmittance spectra of the white-light-emitting SPC$^S$ $_{DPA-C6-MC}$ (DPA/C6/MC solution molar ratio of 8/1/27). Inset: photograph of SPC$^S$ $_{DPA-C6-MC}$ under natural light (left) and UV irradiation (right). **b** CIE chromaticity diagram illustrating the luminescent color of SPC$^S$ $_{DPA-C6-MC}$ (0.30, 0.33). **c** Dissymmetry factor ($g_{lum}$) of SPC$^S$ $_{DPA-C6-MC}$ and SPC$^R$ $_{DPA-C6-MC}$ ($\lambda_{ex}$ = 360 nm). **d** The SPC$^S$ $_{SP}$ film can be used as a smart window (left). The SPC$^S$ film can be cut into a fish model and combined with photofluorochromic SP molecules for anticounterfeiting (right). The images of the fish model before and after UV-365 nm stimulation shows different colors under natural light, and the corresponding reflective color and fluorescence show different intensities under by the L-CPF and R-CPF.

molar ratio 8/1/27). Then, the PBG position of SPC$^S$ film matched with MC emission was selected for soaking in the DPA/C6/MC solution to obtain an SPC$^S$ $_{DPA-C6-MC}$ film. In the fluorescence spectra of the SPC$^S$ $_{DPA-C6-MC}$ film, as shown in Fig. 5a, three emission peaks corresponding to DPA, C6, and MC were observed. As observed in the inset, the SPC$^S$ $_{DPA-C6-MC}$ film showed white emission under 365 nm UV light. Meanwhile, the white-light-emitting SPC$^S$ $_{DPA-C6-MC}$ film possessed a better CIE coordinate value of (0.30, 0.33) as shown in Fig. 5b. As expected, we achieved mirror-imaged circularly polarized light of multi-emission signals from SPC$^S$ $_{DPA-C6-MC}$ and SPC$^R$ $_{DPA-C6-MC}$ (Fig. 5c and Supplementary Fig. 21). Moreover, possible contributions from linear dichroism caused by macroscopic anisotropy during the CPL measurements were eliminated (Supplementary Fig. 21c). The SPC$^S$ $_{DPA-C6-MC}$ and SPC$^R$ $_{DPA-C6-MC}$ films emitted multi-emission CPL with a maximum |$g_{lum}$| up to 0.47. This provides a simple and general approach for fabricating soft films that can emit strong multi-color CPL emission.

Based on the excellent light transmission and large $g_{lum}$ of the chiral SPC film, introducing functional molecules into the unique chiral SPC film can be directly used for smart windows and anticounterfeiting techniques (Fig. 5d). For example, the SP-loaded SPC$^S$ film (SPC$^S$ $_{SP}$) was transparent since SP is colorless. Upon irradiation with UV light, SPC$^S$ $_{SP}$ will change to SPC$^S$ $_{MC}$, which is a purple-color film, and the letters NCNST will be blocked. Finally, SPC$^S$ $_{MC}$ can recover to its initial state under irradiation by visible light (465 nm, LED). This process can be repeated by alternating UV and visible light irradiation (Fig. 5d). Moreover, anticounterfeiting materials were designed by cutting the SP-loaded SPC$^S$ film (fish model, Fig. 5d). Under natural light, the fish model shows light-red colors. When the fish model was exposed to UV light for a few minutes, the fish turned purplish-red with red emission. The color of the fish model returned to the initial state under visible light (465 nm LED light). In addition, the corresponding fish models in each state showed different intensities of the reflective color and

fluorescence when the L-CPF and R-CPF were placed on top of the sample.

## Discussion

In summary, we have described a simple and general approach for constructing CPL-active SPC films. The SPC film with chiral nanopores can endow achiral dyes with a CPL property with both large $g_{lum}$ and high $\Phi$. Chiral SPC films loaded with various dyes always possess high $\Phi$, mainly because the orderly adsorption of dye molecules on the nanopores of the chiral SPC films can avoid severe aggregation of ACQ dyes while restricting molecule rotation of AIE dyes. Based on the excellent CPL performance, the reflection and fluorescence of the dye-loaded chiral SPC film show different intensities under L/R-CPFs, which can be easily distinguished by the naked eye. Together with the good stability and recyclability, these features enable the dye-loaded chiral SPC film to be processible for real-world applications in smart windows and anticounterfeiting. Overall, our findings not only provide a general strategy for constructing excellent CPL materials but also confirm the feasibility of their real practical applications. We hope that this report will provide inspiration to other researchers for designing good CPL materials and promote their application in real life.

## Methods

### Materials

All solvents and chemicals were of reagent quality and used without further purification. A nematic liquid crystal HTG135200 and a chiral dopant $R$5011 (>98%)/$S$5011 (>98%) were provided by Jiangsu Hecheng Display Technology Co., Ltd. 2-Methyl-1,4-phenylene bis(4-((6-(acryloyloxy)hexyl)oxy)benzoate) (C6M, 97%) Trimethylolpropane triacrylate (TMPTA, 95%) and a photoinitiator 2,2-Dimethoxy-2-phenylacetophenone (I-651, 99%) were purchased from Merck. 9,10-Diphenylanthracene (DPA, 98%), Tetraphenylethylene (TPE, >98%) and Coumarin 6 (C6, >98%) were purchased from Shanghai Aladdin

Biochemical Technology Co., Ltd. Perylene (>98%) was purchased from Acros organics. 1',3',3'-Trimethyl-6-nitrospiro(chromene-2,2'-indoline) (SP, 98%) were purchased from TCI Development Co., Ltd. The chemical structures are listed in Supplementary Table 1. Cleaned glass slides were used to prepare liquid crystalline cells.

## Characterizations

UV–vis and fluorescence spectra were recorded on the Hitachi U-3900 spectrophotometer and HORIBA Jobin-Yvon, respectively. The absolute fluorescence quantum yield was measured on HORIBA (FluoroMax+) fluorescence spectrophotometer equipped with an integrating sphere. CD and CPL spectra were measured on JASCO J-1500 and JASCO CPL-200 spectrophotometers, respectively. POM images were recorded on Leica DM2700M upright materials microscope. XRD spectra were measured on the Rigaku D/Max-2500 X-ray diffractometer (Japan) with Cu/K$\alpha$ radiation ($\lambda = 1.5406$ Å). The photoswitching reactions of spiropyran were conducted by irradiation with 465 nm blue LED (HLV2-22BL-3W, CCS Inc.) or hand-held ultraviolet lamp (UV-365 nm, maximum power: 6 W, Shanghai Jiapeng Technology Co., Ltd.). TGA curves were tested on Diamond TG/DTA (5 °C/min). Scanning electron microscopy (SEM) was performed on a Hitachi S-4800 FE-SEM with an accelerating voltage of 10 kV (Prepare cross section samples by hand-tearing the SPC$^S$ and SPC$^S$ $_{SP}$ film). Prior to the measurement, the samples were degassed at 120 °C to remove the physisorbed water and impurities. The pore size distribution was analyzed by ASAP 2460.

## Fabrication of the chiral soft photonic crystal (SPC$^S$ and SPC$^R$) and SPC$^S$ $_{dye}$ (or SPC$^R$ $_{dye}$) films

A CLC mixture containing 10 mg commercial eutectic nematic LC (HTG135200), 0.9 ~ 0.24 mg chiral dopants ($R$5011 or $S$5011), 5 mg reactive mesogen (C6M), 0.78 mg crosslinking agent (TMPTA), and 0.25 mg photoinitiator (I-651) was homogeneously mixed in a chloroform solution. The mixture was heated to 90 °C and filled in LC cell (thickness was 20 μm) by capillary action. Afterward, the sample was exposed to UV irradiation (365 nm) for 30 min to polymerize the orderly arranged C6M and TMPTA; Remove the LC cell. Finally, the SPC$^S$ and SPC$^R$ film was obtained by immersing UV-cured sample in hexane for 12 h to completely clean the N*LC and residual monomer. It was then dried in an oven to vaporize the remnant hexane. The SPC$^S$ and SPC$^R$ film with different PBG was selected to immerse in different dye molecules solution (methanol, [dye] = $5 \times 10^{-4}$ mol l$^{-1}$) for 60 min, rinsed twice in ethanol and oven dry to obtain the SPC$^S$ $_{dye}$ and SPC$^R$ $_{dye}$ film.

## Reporting summary

Further information on research design is available in the Nature Portfolio Reporting Summary linked to this article.

## Data availability

The data that support the findings of this study are available from the corresponding author on request. Source data are provided with this paper.

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

## Acknowledgements

This work was supported by the Beijing Municipal Science and Technology Commission (2212023, JQ21003); the National Key R&D Program of the Ministry of Science and Technology of the People's Republic of China (2021YFA1200303); the Strategic Priority Research Program of Chinese Academy of Sciences (XDB36000000) and the National Natural Science Foundation of China (52173159, 92256304, 22172041, 22205045).

## Author contributions

Y.S. synthesized the samples and carried out all the characterizations. P.D. supervised the work. J.H., C.L., T.Z. and X.J. helped the characterization of the nanostructure and spectral measurements. Y.S. and P.D. analyzed the data and wrote the manuscript. All the authors participated in the discussion and commented on the manuscript.

## Competing interests

The authors declare no competing interests.
