## [Peer Review File · Nature Communications]

Recyclable soft photonic crystal film with overall improved circularly polarized luminescenceEditorial Note: Reviewer #4 was invited to look over the comments from Reviewer #2 and Reviewer #3

REVIEWER COMMENTS

Reviewer #1 (Remarks to the Author):

The authors describe an interesting method for preparation of the high circularity photonic materials with an additional property of recyclability by taking advantage of the chiral nanopore in liquid crystals

Overall this is an interesting paper is suitable for the publication in with minor edits. Provided that the following comments are addressed.

In the introduction, the authors state: .. With regard to real applications of CPL materials, pursuing a large luminescence 34 dissymmetry factor (g_{lum}) is of vital importance but remains a major challenge...." implying that such method has not been developed. In fact, exactly this problem has been extensively discussed in the recent review, that the authors might have missed.

<https://onlinelibrary.wiley.com/doi/full/10.1002/adma.202108431> There is a detailed description of different methods that accomplishing this task. This method is novel, but the proper comparison and description of the mechanisms is still desirable.

Templation of the blue phase is indeed interesting approach. The photonic structures are very useful for obtaining high degree of circularity. However, the they obtain it by using in fact photonic bandgaps originating in contrast of refractive indexes. Scattering properties here are quite critical.

Linear dichroism can also contribute to the effects in Fig 6. It would be reasonable to check for it.

Figure 5 might be more suitable for the SI.

Reviewer #2 (Remarks to the Author):

[Reviewer's Comments]

The manuscript aims to easily prepare high-performance circularly polarized luminescence (CPL) materials using a template film. Although the content is intriguing, including the originality of the template material and the peculiar phenomena in its CPL expression mechanism and luminescence behavior, the manuscript only presents arguments that lack novelty and innovation, making it unsuitable for publication in this prestigious journal. Furthermore, based on the experimental results presented in the manuscript, significant improvements are unlikely, leading to the conclusion that the manuscript should be rejected.

Detailed comments are as follows:

1. The focus of the manuscript is on the mechanism of chirality. However, the manuscript needs to be improved on this point. Although CPL is mentioned in the manuscript as a characteristic chiral property, it arises from the selective reflection properties of the SPC film of the N*-LC structure. The evidence for this idea is that non-CPL is obtained in the blue wavelength region of the white fluorescent film shown in Figure 6 of the manuscript. CPL amplification by selective reflection is a well-known phenomenon. To be accepted by this journal, it needs to be more innovative.

2. The manuscript cites the Cotton effect in the absorption region of the fluorescent substance added to the SPCMC film as the basis for the unique chiral impartation that is not selective reflection. However, this manuscript does not consider the artifacts in the CD or the effects of scattering caused by the SPC film, which retains the N*-LC structure to the extent that it indicates selective reflection. These should of course be taken into account when using anisotropic films or N*-LC structures. Therefore, this Cotton effect is a weak basis for imparting chirality to the fluorophore unless these considerations are taken into account.

3. Moreover, even if the fluorescent substance has inherent chirality, the manuscript does not answer what kind of chirality is imparted. The doped achiral fluorescent substance is assumed to have no aggregation structure or to be in a weakly aggregated state. It would be interesting if chirality could be imparted to fluorophores without forming an aggregate structure and leading to CPL expression. However, this is not demonstrated in this manuscript. As in the work of Coles et al. in the blue phase shown in Ref [33], if a spontaneously oriented liquid crystal material is doped in a porous template, the chiral structure of the template can be transferred to the doped material. However, it is impossible to expect a unique chiral structure based on spontaneous orientation in the general fluorescent materials used in this manuscript. There are no data to prove this. Therefore, it cannot be concluded that a unique chirality is imparted to the fluorescent material.

4. Furthermore, it could be more innovative in terms of its properties as a CPL material. Although the fact that it can maintain a high quantum yield is commendable, the g value shown in this manuscript is about 0.4, which is inferior to typical material systems that use selective reflection.

5. Another noteworthy aspect of this manuscript is the film that combines AIE properties and ACQ suppression, which is a relatively uncommon material and may have some novelty. However, the claim in the manuscript that the mechanism of expression for both AIE characteristics and ACQ suppression is the support of fluorescent substances in the nanopores of the SPC films is rough. The nanopores, at 10 nm, are too large for a single molecule. It is impossible to assume a mechanism to avoid aggregation in ACQ molecules with high intermolecular interactions when a fluorescent molecule enters this very large nanopore.

6. Based on spectroscopic estimates that no aggregation has occurred, it is more reasonable to assume that fluorescent substances are dispersed as single molecules in a state close to dissolution in the SPC film by immersion. In this case, the AIE characteristics resulting from the inhibition of molecular motion are expressed in the film with π -electron interactions, and aggregation of fluorescent molecules is

suppressed. It is evident from the fact that doping a polymer film is a common utilization method for AIE molecules. Since the amount of doped fluorescent substance is not reported in this manuscript, it is unclear whether aggregation is avoided due to low concentration or high quantum yield is obtained even at high concentration. If the former is the case, it is similar to doping a conventional polymer film. If the latter is the case, it demonstrates the innovative nature of the material. However, based on the relationship between concentration and wavelength shown in Supplementary Figure 5, it is likely that the former is the case.

8. More to the point, the application examples shown in the second half of the manuscript have no significant innovation. "White CPL" is a misnomer, since it is not CPL outside the selective reflection region. The cited example of a smart window is simply the function of an optical switching molecule and does not necessarily require the SPC film. There may be other simple and effective anti-counterfeiting methods that do not require the CPL function.

9. Thus, the material and application discussed in this manuscript combine common ideas with no innovative mechanism or material property. It would be interesting if the essential chirality imparting mechanism of achiral fluorescent materials could be demonstrated or if the specificity in AIE and ACQ could be further explored. However, no significant improvements are expected from the current manuscript.

Therefore, it is unlikely to reach the level suitable for publication in this journal and we recommend its rejection.

Reviewer #3 (Remarks to the Author):

The authors report the synthesis of a porous chiral soft photonic crystal in which dyes can migrate. They demonstrate that this approach results in both high quantum efficiencies and strong CPL.

The material is novel and its value is either experimentally demonstrated or motivated. The concept is really nice. Therefore, this study is of interest for a broad audience, which could warrant publication in Nature Communications. I have three major concerns.

1) What is the origin of the porosity? The dimensions of the pores is very important to result in the very high quantum efficiencies, as motivated in the manuscript. How are the dimensions controlled? Why is so much material washed away after the synthesis of the SPC?

2) What is the mechanism of the chirality transfer to the dyes? This is key for the very strong CPL and is not explained in the manuscript. In my opinion, three experimental findings should be taken into account. First, CPL is observed for all dyes, also when the PBG does not match with the bandgap of the dye. The observed CPL therefore originates from the dye. Second, all dyes have approximately the same λ_{glu} , which suggests that the chirality of the SPC is the origin of the chirality of the CPL. Third, the shape of the spectra, including the CD spectra (Figure 10 in SI) is puzzling. If the pores are chirally oriented towards each other and the pores contain the dye molecules, the dye molecules become chirally

oriented (and always in the same way). This could explain the transfer of chirality and the fact that θ_{CD} is always the same. However, this implies that exciton coupling takes place, which is not observed in the spectra. So the crucial aspect of the mechanism behind the general transfer of chirality remains unclear. The authors should provide at least an hypothesis for this mechanism. In addition, I do not really agree that the CD spectra in Figure 10, SI are minor images.

3) The explanation for the high quantum efficiency for all types of dyes is not clearly explained. In fact, it is clearly summarized in the conclusions (which section is, in fact, called "Discussion"), but not in the section "Results".

Some minor comments:

Lines 126-129: Why does the PBG not shift as a result of a differentiating refractive index (washing, removal of material)?

Reviewer #1:

The authors describe an interesting method for preparation of the high circularity photonic materials with an additional property of recyclability by taking advantage of the chiral nanopore in liquid crystals. Overall this is an interesting paper is suitable for the publication in with minor edits. Provided that the following comments are addressed.

Response: Thank you very much for your support and encouragement.

1. In the introduction, the authors state: .. With regard to real applications of CPL materials, pursuing a large luminescence 34 dissymmetry factor (glum) is of vital importance but remains a major challenge....” implying that such method has not been developed. In fact, exactly this problem has been extensively discussed in the recent review, that the authors might have missed. <https://onlinelibrary.wiley.com/doi/full/10.1002/adma.202108431>. There is a detailed description of different methods that accomplishing this task. This method is novel, but the proper comparison and description of the mechanisms is still desirable.

Response: Thank you very much. We have corrected it in the revised manuscript and cited the reference in appropriate position (Ref. 12). The corresponding parts were marked in yellow in the revision manuscript.

2. Templatation of the blue phase is indeed interesting approach. The photonic structures are very useful for obtaining high degree of circularity. However, the they obtain it by using in fact photomic bandgaps originating in contrast of refractive indexes. Scattering properties here are quite critical.

Response: For suppressing the effect of scattering, when we tested all the fluorescence and CPL spectra of chiral SPC films, the excitation light, sample and detector were all fixed on the same axis (Figure R6). Moreover, we have added it in the revised supporting information (Supplementary Fig. 11).

Figure R6 Schematic representation of experimental setup for PL and CPL spectra.

3. Linear dichroism can also contribute to the effects in Fig 6. It would be reasonable to check for it.

Response: Thank you very much for your comments. To avoid the effect of linear dichroism, we conducted CPL spectra testing on the SPC^S_{DPA-C6-MC} by varying the angle of the sample along the direction of incident light propagation. CPL signal intensity of SPC^S_{DPA-C6-MC} was no obvious change, indicating that the contribution of linear dichroism to the CPL signals can be ignored (Figure R7). We have added it in the revised supporting information (Supplementary Fig. 20).

Figure R7 a Schematic representation of experimental setup for CPL spectra. **b** CPL spectra of $\text{SPC}^{\text{S}}_{\text{DPA-C6-MC}}$ and $\text{SPC}^{\text{R}}_{\text{DPA-C6-MC}}$ ($\lambda_{\text{ex}} = 360 \text{ nm}$). **c** The g_{lum} of $\text{SPC}^{\text{S}}_{\text{DPA-C6-MC}}$ by changing the angle of the sample along the direction of incident light propagation.

4. Figure 5 might be more suitable for the SI.

Response: Thank you very much for your comments. We have moved Figure 5 in the revision manuscript to the revised supporting information (Supplementary Fig. 13).

Reviewer #2:

The manuscript aims to easily prepare high-performance circularly polarized luminescence (CPL) materials using a template film. Although the content is intriguing, including the originality of the template material and the peculiar phenomena in its CPL expression mechanism and luminescence behavior, the manuscript only presents arguments that lack novelty and innovation, making it unsuitable for publication in this prestigious journal. Furthermore, based on the experimental results presented in the manuscript, significant improvements are unlikely, leading to the conclusion that the manuscript should be rejected.

Response: In this work, we have demonstrated a novel method for constructing efficient circularly polarized luminescent materials. There are three innovative points: 1) Soft photonic crystal (SPC) film embedded with chiral nanopores can work as a versatile chiral platform for constructing highly efficient CPL materials possessing large luminescence dissymmetry factor, as well as large luminescence quantum yield. 2) This approach is general. All kinds of luminescent dyes, including photochromic dyes, AIE dyes, ACQ dyes, and other normal dyes, can be endowed with excellent CPL emission. 3) This kind solid film enables the wide application. Thus, this work not only streamlines the preparation process of CPL materials but also opens up new possibilities for their practical applications.

1. The focus of the manuscript is on the mechanism of chirality. However, the manuscript needs to be improved on this point. Although CPL is mentioned in the manuscript as a characteristic chiral property, it arises from the selective reflection properties of the SPC film of the N*-LC structure. The evidence for this idea is that non-CPL is obtained in the blue wavelength region of the white fluorescent film shown in Figure 6 of the manuscript. CPL amplification by selective reflection is a well-known phenomenon. To be accepted by this journal, it needs to be more innovative.

Response: Indeed, understanding the mechanism of chiral transfer is of utmost

importance in our work. We have conducted an extensive investigation on this aspect, as outlined in the revised manuscript. One key innovation in our study is the utilization of selective reflection to enhance the circularly polarized luminescence (CPL). This novel approach allows SPC films to serve as a versatile platform for incorporating a wide range of luminescent dyes. By aligning the emission peak of the dyes within or beyond the photonic band gap (PBG), we are able to achieve induced CPL signals. Notably, even in the blue wavelength region of the white fluorescent film, a CPL signal is observed (Figure R8). To ensure the clarity of the CPL signal in this blue region, we have included the corresponding data in the supporting information section of the revised manuscript (Supplementary Fig. 20b).

Figure R8 CPL spectra of SPC^S_{DPA-C6-MC} and SPC^R_{DPA-C6-MC} ($\lambda_{\text{ex}} = 360$ nm).

2. The manuscript cites the Cotton effect in the absorption region of the fluorescent substance added to the SPCMC film as the basis for the unique chiral impartation that is not selective reflection. However, this manuscript does not consider the artifacts in the CD or the effects of scattering caused by the SPC film, which retains the N*-LC structure to the extent that it indicates selective reflection. These should of course be taken into account when using anisotropic films or N*-LC structures. Therefore, this Cotton effect is a weak basis for imparting chirality to the fluorophore unless these considerations are taken into account.

Response: Thank you very much for your comments. The SPC^{S} and SPC^{R} exhibit distinct mirror-image CD signals, which can be attributed to the chiral structures imparted by the chiral liquid crystal (Figure R9a). Notably, a new mirror signal centered at 570 nm (beyond $\text{PBG} = 750$ nm) was observed in the CD spectra of both $\text{SPC}^{\text{S}}_{\text{MC}}$ and $\text{SPC}^{\text{R}}_{\text{MC}}$ samples, strongly suggesting the chiral transfer from chirally imprinted structures to achiral MC dye (Figure R9b). To ensure that the observed signals are not influenced by linear dichroism or scattering, we conducted a rotation angle test on the samples. The g_{CD} value remained unchanged, indicating that the influence of line deviation can be disregarded (Figure R9c, d).

In order to further corroborate the chiral transfer from the chiral-imprinted structures to the achiral dye molecules, we incorporated C6 molecules into the chiral SPC films, which yielded similar results (Figure R10). We have included these additional findings in the revised supporting information (Supplementary Figs. 10, 16).

Figure R9 **a** CD spectra of SPC^{S} and SPC^{R} films. **b** CD spectra of $\text{SPC}^{\text{S}}_{\text{MC}}$ and $\text{SPC}^{\text{R}}_{\text{MC}}$ films. The CD spectra (**c**) and g_{CD} (**d**) of $\text{SPC}^{\text{R}}_{\text{MC}}$ by changing the angle of the sample along the direction of incident light propagation.

Figure R10 CD spectra of SPC^S and SPC^S_{C6}.

3. Moreover, even if the fluorescent substance has inherent chirality, the manuscript does not answer what kind of chirality is imparted. The doped achiral fluorescent substance is assumed to have no aggregation structure or to be in a weakly aggregated state. It would be interesting if chirality could be imparted to fluorophores without forming an aggregate structure and leading to CPL expression. However, this is not demonstrated in this manuscript. As in the work of Coles et al. in the blue phase shown in Ref [33], if a spontaneously oriented liquid crystal material is doped in a porous template, the chiral structure of the template can be transferred to the doped material. However, it is impossible to expect a unique chiral structure based on spontaneous orientation in the general fluorescent materials used in this manuscript. There are no data to prove this. Therefore, it cannot be concluded that a unique chirality is imparted to the fluorescent material.

Response: As you said, it is intriguing how chirally imprinted pores can confer chirality to achiral dye molecules, particularly when they are in a non-aggregated or weakly aggregated state. To get more insights on the chiral transfer from chiral SPC film to achiral dye molecules, we selected C6-loaded chiral SPC^S films as representative

example for elucidation. We conducted investigations on the photophysical properties of SPC^S films by immersing them in a dilute solution of C6 molecules at various concentrations (ranging from 10⁻⁶ to 5 × 10⁻⁵ mol L⁻¹) for 10 min. As the concentration of C6 solution increased, both the absorption and emission of C6 molecules within the SPC^S film also increased. This result suggests a strong correlation between the quantity of embedded dye molecules in the SPC^S film and the concentration of the solution (refer to Figure R11a, b). Interestingly, the position of the absorption spectra of SPC^S C6 did not exhibit a significant shift with an increasing amount of embedded C6 molecules. However, we observed a clear augmentation in the circular dichroism (CD) values of the SPC^S C6 film as the concentration of the C6 molecules increased (Figure R11c, d). Based on the experimental results mentioned above, we concluded that the SPC^S C6 film at low concentration displayed low CD values, possibly due to the limited adsorption of C6 molecules on the chiral nanopores imprint. This limited adsorption resulted in the absence of an orderly arrangement (Figure R11e). Conversely, as the concentration increased, the C6 molecules were more orderly adsorbed onto the chiral nanopores imprint of the SPC^S film, leading to a stronger CD signal. Thus, the chirality of the guest dyes is closely associated with the loading amount, indicating that the induced chirality should be attributed to the supramolecular chirality of the dyes, rather than the selective reflection of the chiral nematic liquid crystal (N*LC).

In summary, the results of our study highlight the ability of chirally imprinted pores to confer chirality to achiral dye molecules, even without strong intermolecular coupling. This discovery holds significant promise and is expected to contribute to the development of novel research directions in the field of CPL-active materials. We have added it in the revised supporting information (Supplementary Figs. 17).

Figure R11 a SPC^S was immersed in solutions of C6 dyes of different concentrations for 10 min and then dried, and image of SPC^S C₆ were photographed the under the nature light (top) and 365 UV light (bottom). The absorption (b) and CD spectra (c) of different SPC^S C₆ film. d Concentrations-dependent CD spectra of SPC^S C₆. e Schematic representation of the adsorption process of dye molecules in chiral nanopores imprint of SPC film. As the embedded dye molecules increase, they adsorb orderly into the chiral nanopores imprint.

4. Furthermore, it could be more innovative in terms of its properties as a CPL material. Although the fact that it can maintain a high quantum yield is commendable, the g value shown in this manuscript is about 0.4, which is inferior to typical material systems that use selective reflection.

Response: Thank you very much for your comments. When considering the g_{lum} value alone, it becomes apparent that the g_{lum} of the SPC film system is lower compared to that of chiral liquid crystal systems. However, when considering the overall performance of the CPL material, the chiral SPC film system outperforms most previously reported CPL systems in terms of the general ease of CPL material preparation, film recyclability, and practical applicability of the film. It is important to note that there is still ample room for further enhancing the g_{lum} value by modifying the composition of the polymeric film, such as selecting appropriate monomers,

crosslinkers, chiral induction reagents, and other factors. This realization serves as a strong motivation for our future exploration of chiral SPC films.

5. Another noteworthy aspect of this manuscript is the film that combines AIE properties and ACQ suppression, which is a relatively uncommon material and may have some novelty. However, the claim in the manuscript that the mechanism of expression for both AIE characteristics and ACQ suppression is the support of fluorescent substances in the nanopores of the SPC films is rough. The nanopores, at 10 nm, are too large for a single molecule. It is impossible to assume a mechanism to avoid aggregation in ACQ molecules with high intermolecular interactions when a fluorescent molecule enters this very large nanopore.

Response: As you pointed out, it is indeed uncommon to find dye molecules that possess both the aggregation-induced emission (AIE) and aggregation-caused quenching (ACQ) effects embedded within the chiral nanopores imprint of an SPC film, all while maintaining a high fluorescence quantum yield. In the original manuscript, we focused on the relationship between the nanopore size of the chiral SPC film and the various dye molecules. In this revised version, the influence of concentration was thoroughly investigated. We have provided experimental data at different concentrations and provided a detailed explanation of the mechanism that results in high fluorescence quantum yield for various dye molecules. This question has been thoroughly addressed in the subsequent questions that you have raised and will be discussed in the following sections. In our response, we have conducted a comprehensive investigation and provided a detailed explanation to shed light on this intriguing phenomenon.

6. Based on spectroscopic estimates that no aggregation has occurred, it is more reasonable to assume that fluorescent substances are dispersed as single molecules in a state close to dissolution in the SPC film by immersion. In this case, the AIE characteristics resulting from the inhibition of molecular motion are expressed in the film with π -electron interactions, and aggregation of fluorescent molecules is suppressed. It is evident from the fact that doping a polymer film is a common

utilization method for AIE molecules. Since the amount of doped fluorescent substance is not reported in this manuscript, it is unclear whether aggregation is avoided due to low concentration or high quantum yield is obtained even at high concentration. If the former is the case, it is similar to doping a conventional polymer film. If the latter is the case, it demonstrates the innovative nature of the material. However, based on the relationship between concentration and wavelength shown in Supplementary Figure 5, it is likely that the former is the case.

Response: Thank you very much for your comments. In order to address this issue comprehensively, we have incorporated additional experiments and included the corresponding data in the revised manuscript and supporting information (Supplementary Figs. 18, 19).

Analysis of the absorption and CD spectra of SPC^S_{C6} in dilute solutions (ranging from 10⁻⁶ to 5 × 10⁻⁵ mol L⁻¹, Figure R11) enables us to infer that the embedded dye molecules are adsorbed in an ordered manner within the chiral nanopores imprint. This ordered adsorption of dye molecules on the chiral nanopores imprint limits their internal rotations, thereby leading to a higher fluorescence quantum yield, particularly in dye molecules with the AIE effect in the SPC^{R or S} films.

To further investigate the fluorescence quantum yield of SPC^{R or S} films following immersion in a highly concentrated dye solution, we chose C6 molecules with the ACQ effect for the experiments. The CD signal of the SPC^S_{C6} film significantly increased with longer immersion time in the supersaturated solutions. When the immersion time reached approximately 14 hours, the dye molecules in the SPC^S film reached saturation (Figure R12a, b). Notably, throughout the entire immersion process, the absorption and emission spectra of SPC^S_{C6} did not show any noticeable shifts (Figure R12c, d). We hypothesize that when the chiral nanopores imprint was completely filled with dye molecules, the solvent's pull prevented significant aggregation, thus keeping them in solution (Figure R13a, b).

To further validate our hypothesis, we examined the dispersion state of the guest dyes by testing the UV-vis and emission spectra. The absorption spectra of the SPC^S_{C6} film in supersaturated solutions for 24 hours displayed a slight red shift compared to

the C6 solution state, indicating minimal aggregation behavior in the film (Figure R13c). Corresponding XRD results also confirmed the absence of severe aggregates of C6 molecules in the SPC^S film (Figure R13d). Similar results were observed for the absorption spectra of perylene molecules in the SPC^S film (Figure R13e). Therefore, it is evident that dye molecules with both ACQ and AIE effects exhibit high fluorescence quantum yields in the chiral SPC film. This can be attributed to the orderly adsorption of dye molecules within the nanopores, which restricts intramolecular rotation. Additionally, the influence of the solvent within the imprinted nanopores prevents significant aggregation among the dye molecules.

In summary, when we immersed the sample in a low-concentration solution, we observed that the CD intensity increased with increasing concentration, indicating the adsorption of dye molecules onto the nanopores. As the immersion time increased in the supersaturated solution, the CD intensity also increased. However, after 14 hours of immersion, the thin film reached a saturation state. At this point, the absorption, emission, and XRD data of the film all indicated a weakly aggregated state of the guest dyes. When we immersed the chiral SPC film, mainly consisting of nanopores with a size distribution of 4-7 nm, in the supersaturated solution, the molecules exhibited a weakly aggregated state within the film. Therefore, we can speculate that when the nanopores of the film are fully occupied by dye molecules and no further adsorption of dye molecules occurs, keeping the dye molecules in the solution, the ACQ molecules in the film exhibit high quantum yield regardless of the concentration and immersion time.

Figure R12 **a** CD spectra of SPC^S film after immersion in supersaturated solutions of C6 dyes for different times. **b** Immerse time-dependent CD signal intensity of SPC^S C₆ film in supersaturated solutions of C6 dyes. **c** Absorption spectra of SPC^S film after immersion in supersaturated solutions of C6 dyes for different times. **d** Emission spectra of SPC^S film after immersion in supersaturated solutions of C6 dyes for 0.25 h and 24 h.

Figure R13 **a, b** Illustration of the adsorption process of dye molecules in chiral nanopores imprint

of the SPC film. **c** Absorption spectra of C6 in solution, SPC^S film and solid state, respectively. **d** XRD spectra of glass, C6, SPC^S, and SPC^S C6. **e** Absorption spectra of perylene in solution ([Perylene] = 2×10^{-5} mol L⁻¹), SPC^S film and solid state, respectively.

8. More to the point, the application examples shown in the second half of the manuscript have no significant innovation. "White CPL" is a misnomer, since it is not CPL outside the selective reflection region. The cited example of a smart window is simply the function of an optical switching molecule and does not necessarily require the SPC film. There may be other simple and effective anti-counterfeiting methods that do not require the CPL function.

Response: The application design of SPC^S or SPC^R film showcases the versatility of its material functions. Despite its simplicity, this design serves as a testament to the practicality of chiral SPC film in the field of CPL. Furthermore, we have made the necessary correction in the revised manuscript, replacing "white CPL" with "multi-color emission CPL." In our pursuit of designing a smart window, we aim to explore additional properties of the SPC^S or SPC^R film, specifically its capacity to effectively transmit light. By investigating and harnessing these properties, we can further enhance the potential applications of SPC^S or SPC^R film beyond CPL, thereby expanding its utility and impact.

9. Thus, the material and application discussed in this manuscript combine common ideas with no innovative mechanism or material property. It would be interesting if the essential chirality imparting mechanism of achiral fluorescent materials could be demonstrated or if the specificity in AIE and ACQ could be further explored. However, no significant improvements are expected from the current manuscript.

Response: Thank you very much for your comments. In this work, we have demonstrated a novel method for constructing efficient circularly polarized luminescent materials. There are three innovative points: 1) Soft photonic crystal (SPC) film embedded with chiral nanopores can work as a versatile chiral platform for constructing

highly efficient CPL materials possessing large luminescence dissymmetry factor, as well as large luminescence quantum yield. 2) This approach is inclusive. All kinds of luminescent dyes, including photochromic dyes, AIE dyes, ACQ dyes, and other normal dyes, can be endowed with excellent CPL emission. 3) This kind solid film enables the wide application. Thus, this work not only streamlines the preparation process of CPL materials but also opens up new possibilities for their practical applications. Furthermore, detailed explanations of the mechanisms behind chiral transfer and high fluorescence quantum yield have been provided in the revised manuscript.

Reviewer #3 (Remarks to the Author):

The authors report the synthesis of a porous chiral soft photonic crystal in which dyes can migrate. They demonstrate that this approach results in both high quantum efficiencies and strong CPL. The material is novel and its value is either experimentally demonstrated or motivated. The concept is really nice. Therefore, this study is of interest for a broad audience, which could warrant publication in Nature Communications. I have three major concerns.

Response: Thank you very much for your support and encouragement.

1) What is the origin of the porosity? The dimensions of the pores is very important to result in the very high quantum efficiencies, as motivated in the manuscript. How are the dimensions controlled? Why is so much material washed away after the synthesis of the SPC?

Response: Thank you very much for your comments. The co-assembly of chiral molecules and nematic liquid crystals (N*LC) leads to the formation of a helical structure (Figure R14, stage I). In this study, we specifically chose polymerizable cross-linkers (TMPTA) and monomer molecules (C6M) and mixed them with chiral liquid crystals. When exposed to 365 nm light, the crosslinker and monomer undergo photopolymerization within the chiral liquid crystal, while the liquid crystal and chiral molecules themselves remain unreactive and can be removed by the solvent. This process leaves behind nanopores imprinted with the texture of the liquid crystal (Figure R14, stage III). Furthermore, we found that the size of the nanopores in thin films can be controlled by adjusting the length of the liquid crystal molecules. Therefore, we propose that modifying the length of these liquid crystal molecules could serve as a means to regulate the nanopore size in chiral SPC films.

Figure R14 (or Figure 1) Preparation of a chiral SPC (SPC^S or R) film and general strategy for obtaining CPL materials. Stage I: Components of an N*LC polymer film. Stage II: Mixture components were exposed to 365 nm for photopolymerization to form N*LC polymer film. Stage III: The N*LC polymer film was placed in hexane to wash out the liquid crystal, chiral dopant and residual monomer. SPC films embedded with chiral nanopores (SPC^S or R) were obtained. Stage IV: Various dye molecules embedded into SPC^S or R films always exhibits large g_{lum} and high Φ after.

2) What is the mechanism of the chirality transfer to the dyes? This is key for the very strong CPL and is not explained in the manuscript. In my opinion, three experimental findings should be taken into account. First, CPL is observed for all dyes, also when the PBG does not match with the bandgap of the dye. The observed CPL therefore originates from the dye. Second, all dyes have approximately the same g_{lum} , which suggests that the chirality of the SPC is the origin of the chirality of the CPL. Third, the shape of the spectra, including the CD spectra (Figure 10 in SI) is puzzling. If the pores are chirally oriented towards each other and the pores contain the dye molecules, the dye molecules become chirally oriented (and always in the same way). This could explain the transfer of chirality and the fact that g_{lum} is always the same. However, this implies that exciton coupling takes place, which is not observed in the spectra. So the

crucial aspect of the mechanism behind the general transfer of chirality remains unclear. The authors should provide at least an hypothesis for this mechanism. In addition, I do not really agree that the CD spectra in Figure 10, SI are minor images.

Response: We appreciate for the valuable suggestions. The unique PBG properties could reflect circularly polarized light with the same handedness and pass through the opposite one, enabling chiral SPC film an excellent matrix for achieving CPL with large g_{lum} value.

The SPC^S and SPC^R show mirror-image CPL signals, indicating the presence of chirality imprinted structures derived from the chiral texture of the liquid crystals (Figure R15a). Intriguingly, in the CD spectrum of SPC^S_{MC} and SPC^R_{MC} , new mirror signal centered at 570 nm emerges, strongly suggesting chiral transfer from the chirality imprinted structures to the achiral MC dye (Figure R15b). To ensure accurate measurements unaffected by linear dichroism, we conducted a rotation angle test on the sample, resulting in an unchanged g_{CD} value and negligible line deviation interference (Figure R15c, d). To further confirm the transfer of chirality from the imprinted structures to the achiral dye molecules, we embedded C6 molecules into the chiral SPC films, which exhibited similar results (Figure R16). We have also included these findings in the revised supporting information (Supplementary Figs. 10, 16).

To gain further insights into the chiral transfer from the chiral SPC film to the achiral dye molecules, we conducted a thorough investigation on C6-embedded SPC^S films. We explored the photophysical properties of the SPC^S film after immersing it in a dilute solution of C6 molecules with varying concentrations (ranging from 10^{-6} to 5×10^{-5} mol L⁻¹) for 10 minutes. As the concentration of the C6 solution increased, both the absorption and emission of C6 molecules within the SPC^S film showed a corresponding increase. This observation suggests a correlation between the amount of embedded dye molecules in the SPC^S film and the concentration of the solution (Figure R17a, b). Importantly, the position of the absorption spectra of SPC^S_{MC} did not exhibit significant shifts with increasing C6 loading amount. However, it was evident that the CD values of the SPC^S_{MC} film increased as the concentration of C6 molecules increased (Figure R17c, d). Based on these results, we postulated that SPC^S_{C6} films with low

concentrations of C6 molecules exhibited low CD values, potentially due to the limited adsorption of C6 molecules onto the chiral nanopore imprints, which resulted in a lack of orderly arrangement (Figure R17e). As the concentration increased, the number of loaded C6 molecules gradually rose, leading to their orderly adsorption onto the chiral nanopore imprints in the SPC^S film, ultimately inducing a stronger CD signal. We have included these additional findings in the revised supporting information (Supplementary Figs. 17).

Figure R15 **a** CD spectra of SPC^S and SPC^R films. **b** CD spectra of SPC^S_{MC} and SPC^R_{MC} films. The CD spectra (**c**) and g_{lum} (**d**) of SPC^R_{MC} by changing the angle of the sample along the direction of incident light propagation.

Figure R16 CD spectra of SPC^{S} and $\text{SPC}^{\text{S}}_{\text{C6}}$.

Figure R17 a) SPC^{S} was immersed in solutions of C6 dyes of different concentrations for 10 min and then dried, and image of $\text{SPC}^{\text{S}}_{\text{C6}}$ were photographed the under the nature light (top) and 365 UV light (bottom). The absorption (b) and CD spectra (c) of different $\text{SPC}^{\text{S}}_{\text{C6}}$ film. d) Concentrations-dependent CD spectra of $\text{SPC}^{\text{S}}_{\text{C6}}$. e) Schematic representation of the adsorption process of dye molecules in chiral nanopores imprint of SPC film. As the embedded dye molecules increase, they adsorb orderly into the chiral nanopores imprint.

3) The explanation for the high quantum efficiency for all types of dyes is not clearly explained. In fact, it is clearly summarized in the conclusions (which section is, in fact,

called “Discussion”), but not in the section “Results”.

Some minor comments: Lines 126-129: Why does the PBG not shift as a result of a differentiating refractive index (washing, removal of material)?

Response: Thank you very much for your comments. To answer this issue, we have added some experiments and placed the corresponding data in the appropriate places in the revised manuscript and supporting information (Supplementary Figs. 18, 19).

Based on the absorption and CD spectra of $\text{SPC}^{\text{S}}_{\text{C6}}$ in the dilute solution (ranging from 10^{-6} to 5×10^{-5} mol L^{-1} , Figure R17), it can be inferred that the embedded dye molecules are adsorbed in an ordered manner within the chiral nanopore imprints. Dye molecules with AIE effect in the $\text{SPC}^{\text{R or S}}$ films has a high quantum yield, primarily because the orderly adsorption of dye molecules onto the chiral nanopore imprints limits their internal rotations.

To further investigate the fluorescence quantum yield of $\text{SPC}^{\text{R or S}}$ films after immersion in highly concentrated dye solution, we selected C6 molecules with an aggregation-caused quenching (ACQ) effect in supersaturated solution for the experiments. The CD signal of $\text{SPC}^{\text{S}}_{\text{C6}}$ film increased with longer immersion time in supersaturated solutions. When the immersion time reached approximately 14 hours, the dye molecules in SPC^{S} film approached saturation (Figure R18a, b). Importantly, the absorption and emission spectra of $\text{SPC}^{\text{S}}_{\text{C6}}$ did not exhibit any shifts throughout the entire immersion process (Figure R18c, d). We hypothesize that when the chiral nanopore imprints are completely filled with dye molecules, the molecules remain in solution due to the solvent's pull, preventing significant aggregation (Figure R19a, b).

To further verify our hypothesis, we investigated the dispersion state of the guest dyes by obtaining UV-vis and emission spectra. The absorption spectra of $\text{SPC}^{\text{S}}_{\text{C6}}$ film in supersaturated solutions for 24 hours showed a slightly red-shift compared to the C6 solution state, indicating minimal aggregation behavior in the film (Figure R19c). The corresponding XRD results also confirmed that the C6 molecules did not form severe aggregates in the SPC^{S} film (Figure R19d). Similar results were observed for the absorption spectra of perylene molecules in the SPC^{S} film have (Figure R19e). Therefore, based on these findings, it becomes evident why dye molecules with both

ACQ and AIE effects exhibit high fluorescence quantum yields in the chiral SPC film. This is attributed to the orderly adsorption of dye molecules within the nanopores, which restricts intramolecular rotation. Simultaneously, the solvent's influence on the dye molecules within the imprinted nanopores prevents significant aggregation. Additionally, it is worth noting that after the film is washed to remove the liquid crystal, the photonic bandgap (PBG) will be blue-shifted compared to its state before washing (Figure R20).

Figure R18 a CD spectra of SPC^S film after immersion in supersaturated solutions of C6 dyes for different times. b Immerse time-dependent CD signal intensity of SPC^S C₆ film at 460 nm in supersaturated solutions of C6 dyes. c Absorption spectra of SPC^S film after immersion in supersaturated solutions of C6 dyes for different times. d Emission spectra of SPC^S film after immersion in supersaturated solutions of C6 dyes for 0.25 h and 24 h.

Figure R19 **a, b** Illustration of the adsorption process of dye molecules in chiral nanopores imprint of the SPC film. **c** Absorption spectra of C6 in solution, SPC^S film and solid state, respectively. **d** XRD spectra of glass, C6, SPC^S, and SPC^S C6. **e** Absorption spectra of perylene in solution ([Perylene] = 2×10^{-5} mol L⁻¹), SPC^S film and solid state, respectively.

Figure R20 Transmission spectra of the SPCS film production process include before polymerization, after polymerization (UV-365 nm) and remove liquid crystal (the picture in inset).

REVIEWER COMMENTS

Reviewer #1 (Remarks to the Author):

Looking good.

The authors addressed my comments fully.

Reviewer #4 (Remarks to the Author):

The suggestions by these two reviewers have been considered. However, the responses seem to not sufficient. On the first hand, the strong light scattering drives the high quantum yield, which can be clearly identified in the UV-vis spectra and SEM images. The defects can be further identified by taking POM images in the reflection mode. On the second hand, it is hard to believe that the FL molecules can not penetrate into the LC polymer network. Due to π - π interactions between C6M and FL molecules, the FL molecules should partially embed into the LC polymer network. This manuscript can be accepted after minor revision.

Reviewer #1:

Looking good. The authors addressed my comments fully.

Response: Thank you very much for your support and encouragement.

Reviewer #4:

The suggestions by these two reviewers have been considered. However, the responses seem to not sufficient. On the first hand, the strong light scattering drives the high quantum yield, which can be clearly identified in the UV-vis spectra and SEM images. The defects can be further identified by taking POM images in the reflection mode. On the second hand, it is hard to believe that the FL molecules can not penetrate into the LC polymer network. Due to pi-pi interactions between C6M and FL molecules, the FL molecules should partially embed into the LC polymer network. This manuscript can be accepted after minor revision.

Response: Thank you very much for your comments. UV-vis spectra reveal the presence of scattering, but we adopt the absolute fluorescence quantum yield test using a blank film as a reference to eliminate the interference of scattering on the fluorescence quantum yield. Additionally, the excitation light is single-wavelength with a very narrow bandwidth, which eliminates any scattering interference.

To address the permeation effects of dye molecules within polymers network, we fabricated polymer film without chiral nanopore as a control experiment. From Figure R1a, we observe that the film with chiral nanopore, after being immersed in a methanol solution containing spiropyran molecules for 6 hours, exhibits minimal fluorescence intensity. In contrast, the SPC film displays strong fluorescence under the same immersion conditions. Corresponding fluorescence spectra shows similar results (Figure R1b). Consequently, we infer that molecular permeation within the polymer membrane is limited and is not the primary factor under consideration.

Figure R1. Polymer film without chiral nanopores made from monomers (C6M), initiators (I-651) and crosslinking agents (TMPTA). (a) Photographs of SPC film (left) and polymer film without chiral nanopores (right) soaked in spiropyran methanol solution for 6 hours. (b) Corresponding fluorescent spectra.

REVIEWERS' COMMENTS

Reviewer #4 (Remarks to the Author):

It is acceptable and suitable to be published as it was.